# Evaluating Geospatial Data Adequacy for Integrated Risk Assessments: A Malaria Risk Use Case

Linda Petutschnig [1,2,*], Thomas Clemen [2], E. Sophia Klaußner [1], Ulfia Clemen [2] and Stefan Lang [1]

1 Christian Doppler Laboratory for Geospatial and EO-Based Humanitarian Technologies, Department of Geoinformatics—Z_GIS, Paris Lodron University of Salzburg, 5020 Salzburg, Austria; stefan.lang@plus.ac.at (S.L.)
2 Department of Computer Science, Hamburg University of Applied Sciences, Berliner Tor 7, 20099 Hamburg, Germany; thomas.clemen@haw-hamburg.de (T.C.); ulfia.clemen@haw-hamburg.de (U.C.)
* Correspondence: linda.petutschnig@plus.ac.at

**Abstract:** International policy and humanitarian guidance emphasize the need for precise, subnational malaria risk assessments with cross-regional comparability. Spatially explicit indicator-based assessments can support humanitarian aid organizations in identifying and localizing vulnerable populations for scaling resources and prioritizing aid delivery. However, the reliability of these assessments is often uncertain due to data quality issues. This article introduces a data evaluation framework to assist risk modelers in evaluating data adequacy. We operationalize the concept of "data adequacy" by considering "quality by design" (suitability) and "quality of conformance" (reliability). Based on a use case we developed in collaboration with Médecins Sans Frontières, we assessed data sources popular in spatial malaria risk assessments and related domains, including data from the Malaria Atlas Project, a healthcare facility database, WorldPop population counts, Climate Hazards group Infrared Precipitation with Stations (CHIRPS) precipitation estimates, European Centre for Medium-Range Weather Forecasts (ECMWF) precipitation forecast, and Armed Conflict Location and Event Data Project (ACLED) conflict events data. Our findings indicate that data availability is generally not a bottleneck, and data producers effectively communicate contextual information pertaining to sources, methodology, limitations and uncertainties. However, determining such data's adequacy definitively for supporting humanitarian intervention planning remains challenging due to potential inaccuracies, incompleteness or outdatedness that are difficult to quantify. Nevertheless, the data hold value for awareness raising, advocacy and recognizing trends and patterns valuable for humanitarian contexts. We contribute a domain-agnostic, systematic approach to geodata adequacy evaluation, with the aim of enhancing geospatial risk assessments, facilitating evidence-based decisions.

**Keywords:** geospatial data; data quality; risk assessment; malaria risk; spatial indicators

## 1. Introduction

The growing availability of freely accessible geospatial data with continental or global coverage steadily expands the range of possible applications for integrated, indicator-based risk assessments [1].

Geospatial risk modelers have access to a multitude of diverse data sources, including satellite measurements and their derivatives, modeled and surveyed data, registry data, multi-source data, volunteered geographical data and data from social media or other involuntary sources. The plethora of available data necessitates a systematic approach to evaluate the adequacy of a given dataset for a specific research question. Therefore, this paper presents a data evaluation framework, designed to be domain-agnostic within geospatial indicator-based risk assessments, to facilitate comparison and assessment of diverse data sources systematically.

In this article, we demonstrate the application of our evaluation framework based on a malaria risk assessment, which we conducted in collaboration with stakeholders

from Médecins Sans Frontières (MSF). The use case showcases the practical utility of the framework and highlights its potential to enhance the reliability of spatial risk assessments.

## 2. Problem Statement

Integrating data from various sources into a comprehensive risk assessment is a common approach to understanding risks with multiple drivers, and is used, for example, in public health contexts such as vector-borne diseases [2–6] or humanitarian research [7–10]. However, in regions with limited health surveillance and general population data, traditional quantitative validation of the final risk score and the reliability of the overall assessment can be challenging [11]. This scarcity is especially prevalent in the WHO Africa region [12]. In contexts defined by data scarcity, the validity of an assessment inherently depends on the conceptual framing of the risk and the adequacy of the used data, a concept referred to as process validity [13]. Process validity involves defining a clear conceptual framework, identifying data sources and associated assumptions, and ensuring transparency in the choices of indicators, sub-indices and aggregation functions [13–15]. This paper focuses on the part of process validity that is concerned with identifying reliable and suitable data, to which we will refer as data adequacy.

In the current era, where institutions and researchers commit to adhering to FAIR (findable, accessible, interoperable, reusable) data sharing principles as part of the wider movement to create research that is replicable and reproducible (R&R) [16], the individual reusing a dataset does not need to possess a comprehensive understanding of underlying methodologies and constraints. It is technically simple to integrate the data into their model or assessment and obtain seemingly conclusive results. This also applies to integrated risk assessments that often rely on open geospatial data, but a thorough assessment of the adequacy of the used data is often not explicitly provided. This can be problematic, as limitations such as incompleteness, inaccuracy or outdatedness can affect the reliability of the findings and conclusions drawn.

Sensitivity on the (in-)adequacy of data is currently driven by the machine- and deep-learning (DL) community, raising awareness for the potential of models trained on biased or incomplete data to exhibit, for example, discrimination against underrepresented groups [17,18]. However, potential harm caused by inadequate data is relevant in all data-driven applications, in particular those that impact humans [19,20], including risk assessments. In the geospatial community, the recent wave of DL-derived insights have triggered criticism pertaining to their varying quality [21–23]. Ref. [24] found that datasets may assert representativeness, when in reality, they only capture a subset of the population, such as social media users or the heads of households. This misperception can result in interventions being designed for only a fraction of the true population, rather than its entirety. Consequently, while data generated through DL methodologies are of great potential, they may have limitations for specific use cases that require careful consideration [25]. If, however, adequate data are used and the indicators are reliable and informative, the assessment can become a crucial aid in resource allocation, and may even serve as an early warning tool for anticipatory action [26–28].

### 2.1. Evaluating Data Adequacy

While the general commitment in the geospatial community to adopt FAIR data sharing principles as a part of the general R&R principles ensures that someone *can* reuse a dataset, a standardized set of metadata that aids the decision of whether a dataset *should* be reused is not yet fully mature. However, various ongoing efforts aim to develop common metadata standards to enhance the user's ability to make an informed and confident decision of whether or not to use a given dataset for their purposes [29,30]. Often, guidance documents and methodological explanations are published, which detail used methodologies and the resulting strengths and limitations [12,31–36]. Among the users of Essential Climate Variable (ECV) data, ref. [29] identified a strong need for guidance on data products and their quality metrics, traceability chains for product algorithms and inter-comparisons

of datasets with similar aims. Simultaneously, the Humanitarian Data Exchange (HDX) platform is currently developing strategies to inform about dataset adequacies in humanitarian contexts, where often timeliness and accuracy must be weighed against each other [37]. Riedler and Lang [30] developed a data evaluation framework that supports evaluating the adequacy of a satellite image to be used as the basis for a given information layer. However, currently, no prevailing framework is recognized among data users developing geospatial risk assessments. Consequently, adequacy is often determined unsystematically, risking a tendency to rely on familiar or frequently used datasets only. Another current challenge users face is the time-consuming need to gather information from various sources, such as different websites, journal publications and methodology reports, as these details are typically not directly or consistently part of the dataset metadata.

### 2.2. Aim

In this article, we present a data evaluation framework that is tailored towards geospatial data, while the individual evaluation criteria are designed to be domain agnostic. We demonstrate its usability on a use case of an indicator-based malaria risk assessment that was developed in partnership with stakeholders from MSF. MSF is a global humanitarian organization that provides essential medical assistance to people affected by conflict, epidemics and natural disasters. The manifold benefits of engaging stakeholders in risk assessment development are well documented [38–40]. The evaluation criteria were discussed with and deemed relevant by the different stakeholders. We applied the framework on various datasets to gain insight whether they would exhibit an adequate quality to support operational intervention planning. Organizations working in the health-provisioning humanitarian domain need to be able to quickly respond to various disasters and circumstances. Therefore, having a robust overview of the available data's qualities is paramount to offer the best possible support to humans in need of assistance.

Our target audience includes both geospatial experts who are involved in mapping and providing information services, as well as domain experts. From any user's perspective, the framework may aid in planning a systematic approach to data adequacy evaluation. From a provider's perspective, including the creators of the datasets themselves or any derived product of them, it shows real-world applications in which their data are being considered for use. The framework lays open the characteristics that are relevant from the user's perspective.

## 3. Background and Methods

### 3.1. Development of the Geodata Evaluation Framework

Our geodata evaluation framework operationalizes the concept of adequacy based on criteria selected from Nightingale [29,41], Riedler and Lang [30], GRID3 [42] and the Dublin Core Metadata Standards [43].

We define adequacy as follows:

$$\text{Adequacy (fitness for purpose)} = \text{Suitability (quality by design)} \times \text{Reliability} \\ \text{(quality of conformance)}$$

Adequacy describes the fitness of a dataset for a given purpose, which is determined by two factors. Suitability or "quality by design" refers to the inherent and intentional quality constraints in the production of data (e.g., spatial resolution of a satellite image, raster cell size of a population dataset, etc.). Reliability or "quality of conformance" focuses on accuracy and completeness in representing a certain geographical area, the population under concern or any other phenomenon. For both factors, the extent to which the data aligns with the particular needs of the use case must be evaluated (definitions based on [30,44]).

In addition, we utilize a range of general metadata to describe the dataset, its capabilities and the data producer. All criteria are listed and described in Table 1. Assessing a dataset's "quality of conformance" is comparatively more difficult, as "quality by design" criteria are generally known a priori and are well documented (e.g., the spatial coverage

of a dataset). "Conformance" may require the data creator's judgment, particularly when comparing data with a reliable validation source is not possible. Therefore, the framework determines "quality of conformance" through the availability of documentation regarding data and the methodologies and sources used. The evaluation supports in qualitatively assessing data adequacy, but it does not provide a means for quantitative comparisons between datasets that use scoring or ranking systems. While having all the information in one place does not enable a definitive and binary decision, it does facilitate a comprehensive overview of potential limitations and opportunities that should be considered when deciding whether to use a dataset.

**Table 1.** List of data evaluation criteria. The criteria are used to operationalize the concept of adequacy.

| Criteria | | |
| --- | --- | --- |
| **Quality by Design Criteria** | | |
| Coverage | Spatial coverage | The geographical extent covered by the resource |
| | Temporal extent | The earliest and latest times covered by the resource |
| Resolution | Spatial resolution | The level of detail in the resource's spatial representation |
| | Temporal resolution | The time interval represented by the resource, e.g., daily, monthly |
| Quality of conformance criteria | | |
| Methodology | Comprehensive method documentation | Availability of a detailed explanation of the resource's content and origin by its creators |
| | Short and easy user guide | Availability of a brief overview of the data's content and origin by its creators |
| | Availability of code | Availability of the model's source code if applicable |
| Traceability of source data | Input/ancillary data | Traceability of datasets used as input or support for modeling resources |
| Strengths and limitations of data | Limitations | Limitations of the resource as stated by its creators |
| | Strengths | Strengths of the resource as stated by its creators |
| Uncertainty characterization | Uncertainty characterization method | The approach used to express uncertainty in the resource |
| | Sources of uncertainty | Origins of uncertainty in the resource's data |
| | Temporal stability uncertainty | Addresses comparability issues due to changes in methodology over time |
| | Geolocation accuracy | Precision of the resource's spatial accuracy |
| Validation | Validation method | The method employed to validate modeled resources |
| Intercomparison | Description of intercomparison activities | Availability of a document that compares resources with similar aims |
| General metadata | | |
| Dataset | Title | A name given to the resource |
| | Identifier | An unambiguous reference to the resource |
| | Date published/produced | A time associated with an event in the resource's lifecycle |
| | Language | The language of the resource |
| | Description | A description of the resource's content |
| | Creator | The main entity responsible for creating the resource |
| | Citation | An official reference provided by creators/publishers |
| | Associated project | The project name where the resource was or is being developed |
| | Publisher | An entity responsible for making the resource available |
| Capabilities | Access options | Methods available to access the resource, such as web scraping |
| | Login required | Indicates if access to the resource requires registration or access key |
| | Format | The file format(s) in which the resource is available |
| | Rights | Information about rights associated with the resource |
| Reputation of data producer | Background of data producer | A brief description of the data producer |

### 3.2. Use Case Scoping and Indicators

Our malaria risk use case aimed at identifying locations of populations in possible need of malaria-related healthcare assistance in a transboundary region encompassing Uganda, Rwanda, Burundi and the provinces of Ituri, North Kivu and South Kivu in the Democratic Republic of the Congo (DRC) (see Figure 1). This region of interest (ROI) as well as the critical malaria risk-related information needs were identified through a series of online meetings with MSF stakeholders working in epidemiology. The defined target was to identify regions exhibiting "emergency settings", which were jointly defined as locations experiencing an interplay of violence, forced migration and limited healthcare, which are known to be prone to malaria outbreaks, and which are expected to experience above-average precipitation during the upcoming malaria season. The last point adds a forecasting component to the assessment, enabling proactive intervention planning ahead of the peak malaria transmission season. The use case is centered around the year 2020.

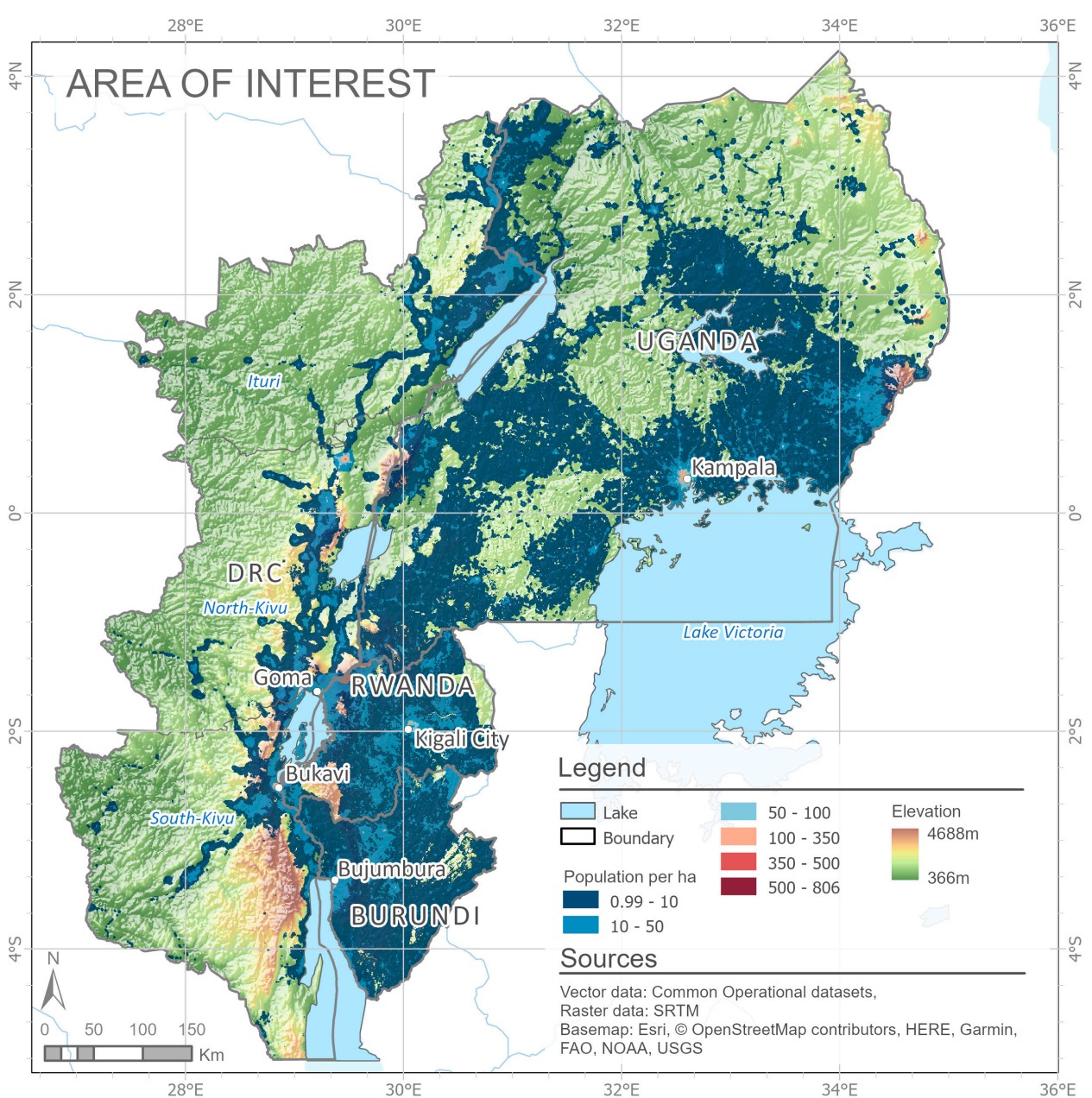

**Figure 1.** The ROI for the case study. The region exhibits significant variations in both topographic features and population distribution.

The indicators we settled for were the following:

1. The seasonal malaria pattern during a normal year;
2. The climate in the upcoming months being particularly conductive to mosquito breeding, i.e., expectations of above-average precipitation;
3. Limited access to healthcare;
4. Ongoing conflicts.

Knowing the spatial variation of these factors would aid in intervention planning, including health post distribution, bednet distribution, indoor residual spraying and awareness-raising campaigns.

### 3.3. Malaria in the Region of Interest

Past efforts to improve access to malaria treatment and prevention have led to a significant reduction in malaria morbidity and mortality in the ROI and beyond [12,45]. However, sustaining this progress remains challenging, e.g., due to ongoing conflict and displacement, political instability, weak health systems and limited healthcare access in rural and remote areas [12,46]. Furthermore, the emergence of drug-resistant malaria strains and re-emergence of the disease in previously controlled areas are of growing concern [47]. Due to this spatially fragmented risk situation, malaria control activities have shifted from (inter-) national interventions to more targeted sub-national interventions [45,48].

The INFORM Epidemics Risk Index 2020 classifies DRC and Burundi as "Very high risk" countries in terms of epidemics risk. Uganda is categorized as "High risk", and Rwanda as "Medium risk" [49]. MSF provides essential medical care to individuals impacted by conflict and displacement in the ROI. In 2022, MSF was active in in all of the countries except for Rwanda, effectively treating 757,800 malaria cases in DRC and 571,000 in Burundi [50].

### 3.4. Applying the Framework to the Use Case

For each indicator, we selected geodata sources that are common choices in spatial malaria risk assessments and related domains; see, e.g., [51] (Table 2). Additionally, a prerequisite for selection was that the data had to be openly accessible for research purposes and encompass the entire ROI. We applied the developed framework to each of the datasets to evaluate their adequacy for our use case. The following sections provide a brief overview of the evaluated data sources and the indicators we calculated based on them. The complete evaluation details are provided in Annex I (Supplementary Materials).

**Table 2.** The left column shows the identified risk drivers, the right column shows the data sources selected to address them.

| Indicator | Evaluated Data Source |
| --- | --- |
| The seasonal malaria pattern during a normal year | "Number of newly diagnosed Plasmodium falciparum cases per 1000 population, on a given year" datasets from the Malaria Atlas Project (MAP) [52] |
| The climate in the upcoming months being particularly conductive to mosquito breeding, i.e., expectations of above-average precipitation | "Total precipitation anomalous rate of accumulation" from the "Seasonal forecast anomalies on single levels" dataset [53] |
| | 30 years of monthly "CHIRPS—Rainfall Estimates from Rain Gauge and Satellite Observations" precipitation estimates [32] |
| Limited access to healthcare | "Walking-only Travel Time to Nearest Healthcare Facility without Access to Motorized Transport" from the MAP [31,34] |
| | "Population Counts—Unconstrained individual countries 2020 UN adjusted, 1 km resolution" by WorldPop [33] |
| Ongoing conflicts | Armed Conflict Location and Event Data (ACLED 2023) |

### 3.4.1. The Seasonal Malaria Pattern

The dataset we chose to represent malaria incidence was created by the Malaria Atlas Project (MAP) [12,54]. We chose the dataset that quantifies the incidence of Plasmodium falciparum (Pf) malaria, which is the predominant parasite in sub-Saharan Africa [55]. The MAP is a renowned academic group, offering geospatially and temporally disaggregated estimations of malaria incidence and mortality. However, the yearly temporal resolution of the dataset is not suitable for the determination of seasonality. This limitation was addressed by reformulating the indicator to emphasize development of yearly average malaria incidence over time. A visual analysis indicated a general trend of a substantial decrease in malaria incidence from 2000 until approximately 2013 across the majority of locations in the ROI, followed by a resurgence in numbers since 2013 (see Figure 2). This led us to calculate the final indicator based on the percentage change in malaria incidence in each location between 2013 and 2020.

Number of newly diagnosed Plasmodium falciparum cases per 1000 population, on a given year

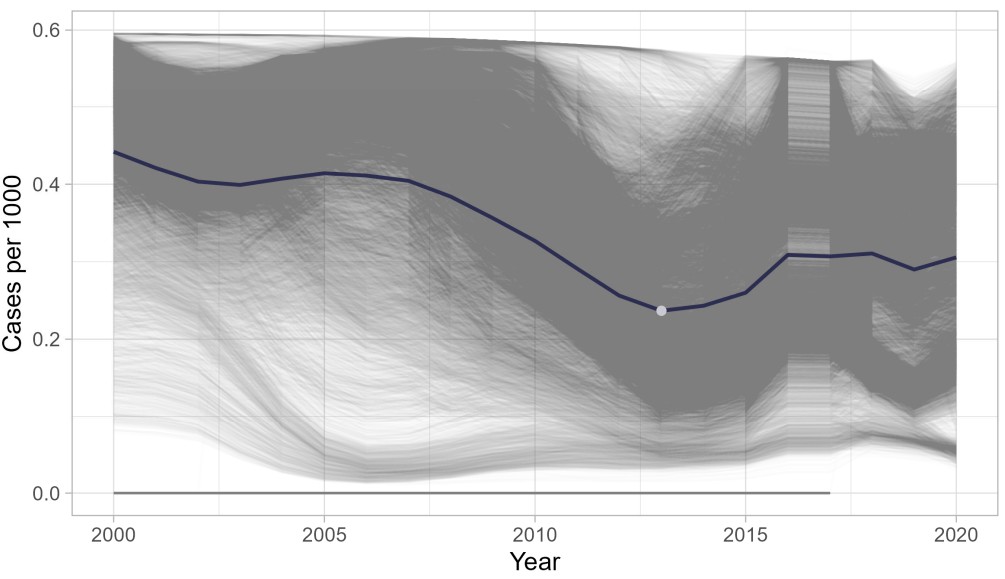

**Figure 2.** Shows the development of diagnosed Pf cases. Each line represents one 5 × 5 km grid cell. The bright point marks the year 2013 after which a trend reversal is noticeable. Source data: Malaria Atlas Project.

### 3.4.2. Precipitation Being Conductive to Mosquito Breeding

Informed by the stakeholders, it was conveyed that above-average precipitation either during or prior to the rainy season serves as a current indicator for anticipating a strong malaria season, as it promotes mosquito breeding conditions. To operationalize this insight, understanding the timing of the rainy season in the ROI was essential. The diverse topography and associated differences in precipitation regimes disqualified hard-coding season boundaries. Instead, we conducted an analysis of 30 years of monthly CHIRPS precipitation estimates [32], spanning 1991–2020, with a resolution of 0.05° × 0.05°, in order to identify year-round monthly precipitation patterns across all locations (see Figure 3).

To determine the upcoming potential for higher malaria occurrences, we utilized the "Total precipitation anomalous rate of accumulation" from the "Seasonal forecast anomalies on single levels" dataset accessible through the Copernicus Climate Change Service (C3S) Climate Data Store (CDS) [53] (see Figure 4). The data are based on the SEAS5 real-time seasonal forecast system run by the European Centre for Medium-Range Weather Forecasts (ECMWF) [36].

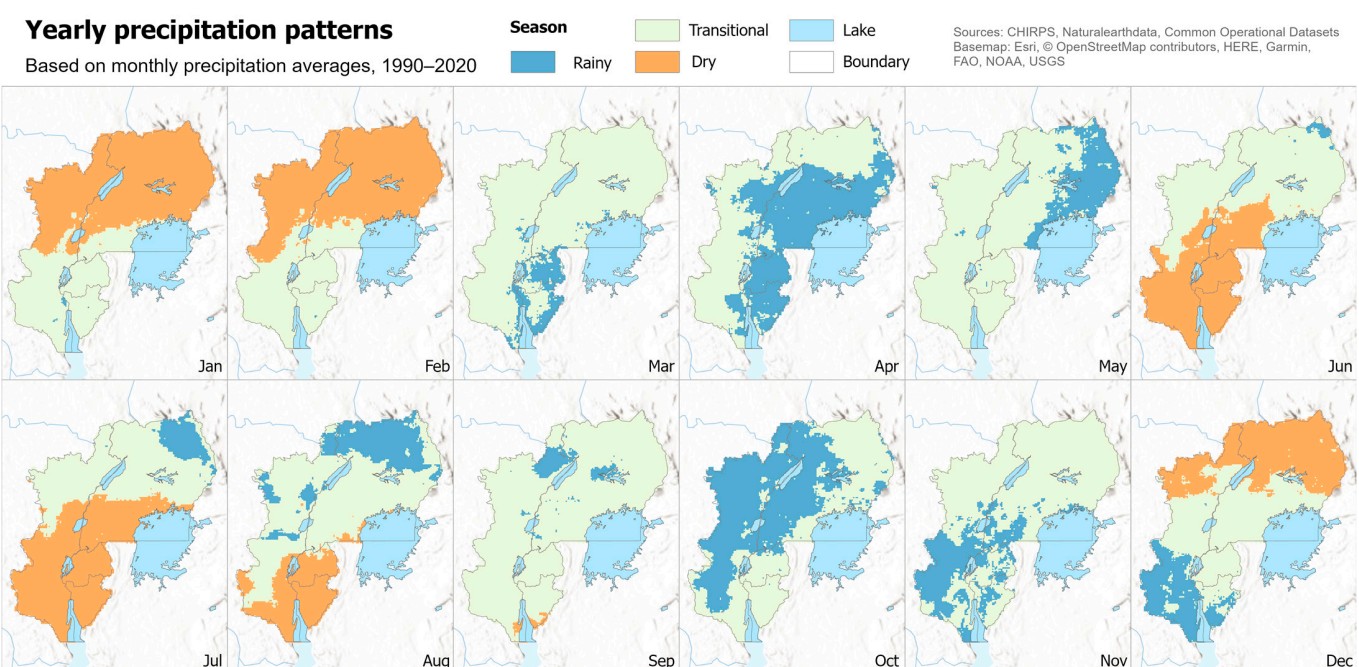

**Figure 3.** For each location, the months were categorized into three seasons based on their 30-year average rainfall: dry, transitional and rainy. To allocate months to the seasons, the annual precipitation at each location was divided by 12 to establish the average monthly rainfall without seasonal variation. Months with an average rainfall exceeding one or more standard deviations above this average were labeled as rainy season and assigned 1 point. Conversely, months with an average rainfall of one standard deviation or more below this average were classified as dry season and assigned −1 point. Months falling in between were designated as transitional season and assigned 0 points.

To combine both information layers, the points acquired for each location were summarized. The assessed timeframe covered February to July 2020. The results, comprising six layers, one for each month, were then summarized into a single final layer for easier integration with the rest of the assessment.

### 3.4.3. Limited Access to Healthcare

To highlight areas with limited access to healthcare, we developed two related indicators. The first assesses the average walking time to the nearest healthcare facility. We used the "Walking-only Travel Time to Nearest Healthcare Facility without Access to Motorized Transport" dataset, provided by the MAP [52]. The healthcare facilities' locations underlying this dataset were initially compiled by [34]. The accessibility information is derived from a friction surface, available globally, that enables calculation of travel times (by foot) from and to all locations [56]. This surface, in combination with the healthcare facility locations, was eventually used by Weiss et al. [31] to model the healthcare accessibility layer.

The second healthcare accessibility indicator estimates the number of individuals expected to seek care at a specific facility, assuming they would choose the facility that is easiest to reach. To achieve this, the previously described accessibility layer and healthcare facility layer were combined with population counts from WorldPop (unconstrained individual countries 2020, UN adjusted, 1 km resolution) [57]. The WorldPop data were selected for their global coverage, spatial resolution and suitability in cases where census data is of poor quality, outdated or non-existent [33]. We applied the "Allocated cost" algorithm, as implemented by SAGA GIS [58], to calculate service areas around each facility, for which we then summed the population counts.

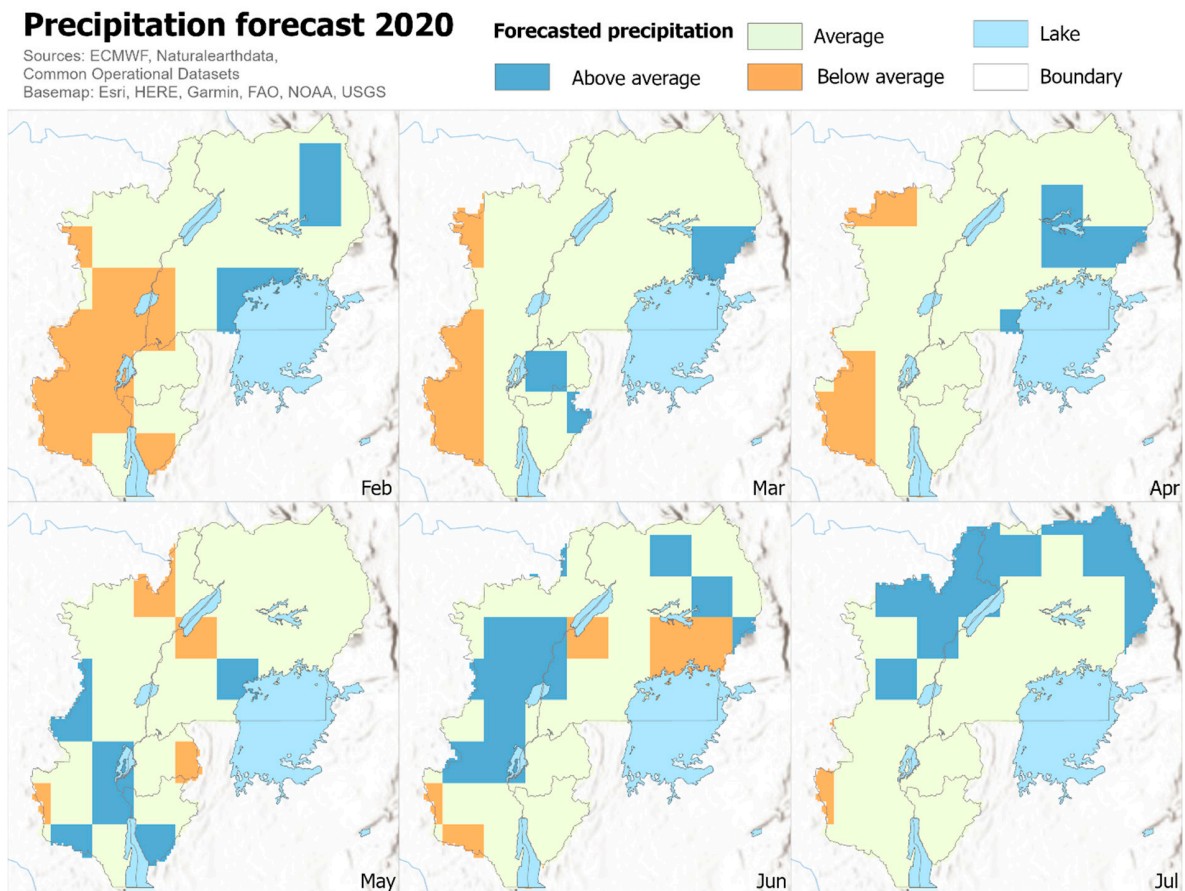

**Figure 4.** In the ECMWF precipitation forecast data, negative values denote below-average anticipated precipitation, while positive values indicate above-average expectations. To identify areas with expected above-average precipitation relative to the anomalies projected throughout the ROI, we first calculated the global mean of the forecast data for the ROI. Locations with expected anomalies exceeding one standard deviation from the global mean were assigned 1 point, whereas areas with expected anomalies one standard deviation or more below the global mean received −1 points. Locations that fall in between were assigned 0 points.

### 3.4.4. Ongoing Conflicts

To determine ongoing conflicts, we utilized the Armed Conflict Locations and Events Database (ACLED). We used three years of events (2017–2019, divided into 3-month intervals) in conjunction with a space-time hotspot detection algorithm [59] to classify the ROI into regions of persistent, intermittent, emerging and former hotspots of conflict, and regions with no discernible conflict pattern.

### 3.5. Data Integration

Once adequacy was evaluated and indicators were formed, the different datasets were integrated using a hexagonal discrete global grid system (DGGS) with target hexagons of 252 km² [60]. DGGSs have the ability to provide a consistent spatial framework, ensuring uniform data representation and analysis [61–63]. While it is common to aggregate indicators to administrative units, our use case aimed to maintain more spatial detail by displaying data on relatively small spatial units that can later be aggregated based on the variability of the phenomenon under concern [64].

This research builds the foundation to subsequent assessment steps such as weighting indicators and performing spatial clustering. These subsequent steps are not part of this article.

*3.6. Code Availability*

The developed analysis workflow, including data access, processing and analysis, were coded in R, with the aim to make the workflow automated and reproducible. All used scripts can be found on GitHub [65].

## 4. Results

In the results section, we analyzed the extent to which each data source aligns with our use case (adequacy), as evaluated through our framework. We concentrated on a limited set of criteria, as a comprehensive analysis of all criteria falls beyond the scope of this article. All of the evaluated datasets are described first by their quality by design features, and then by their quality of conformance.

*4.1. Percentage Change in Malaria between 2013 and 2020*

Data: Number of newly diagnosed Plasmodium falciparum cases per 1000 population in a given year (Malaria Atlas Project).

### 4.1.1. Quality by Design

To create these data, the MAP applied geostatistical models to malaria parasite survey points and routine surveillance reports, along with comprehensive geospatial covariates characterizing Anopheles mosquito habitats [12].

The dataset's design aligns well with the objectives of the use case due to its global coverage and spatial resolution of 0.05° (circa 5 km), which renders it sufficiently granular for interventions targeted at the local scale. Furthermore, the temporal coverage of two decades (2000–2020) aligns with the specifics of our use case.

Figure 5 shows a map of the MAP input data (left) and our indicator (right).

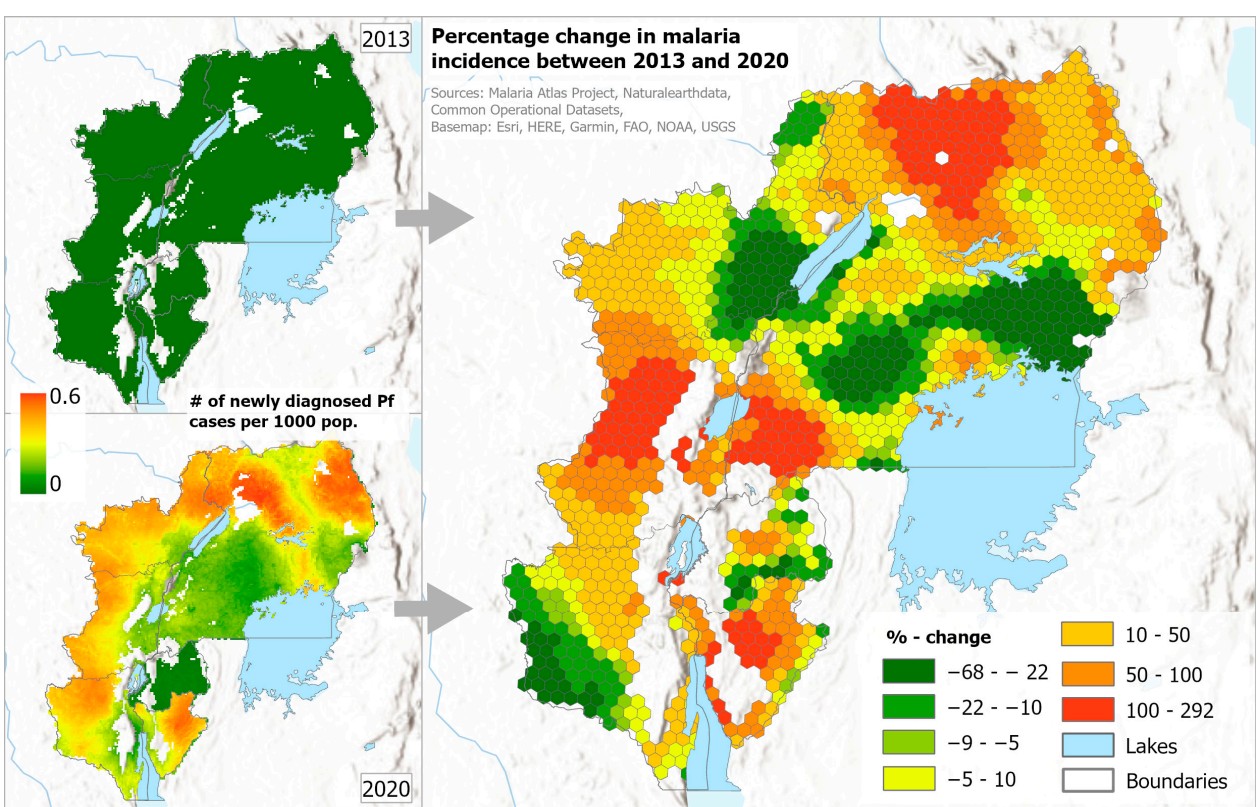

**Figure 5.** The distribution of Pf malaria in 2013 and 2020 (**left**), and the percentage change of the two years integrated into our grid (**right**).

### 4.1.2. Quality of Conformance

The methodology and input data for the MAP data are extensively documented (see Table 3). Strengths and limitations, sources of uncertainty and attempts to validate the data are described in [12]. However, the methodology employed contains uncertainties that are challenging to quantify. This includes, for example, the fact that there is no independent source that the results could be compared to, as the only other two global malaria burden estimates, the World Malaria Report and Global Burden of Diseases (GBD) studies, are in part informed by the MAP data [12]. Furthermore, the spatial disaggregation technique relies on various input data that come with their own inaccuracies and uncertainties, which are propagated into the final product. These uncertainties provoke the question of how accurate the data are at the cell level. However, given that our target spatial scale is a hexagonal grid of 252 km$^2$ and the exact case numbers are less interesting than the general trend, we deemed the source insightful for revealing patterns of change, especially in areas where adjacent hexagons exhibit the same trend.

**Table 3.** The information presented in this table all comes from [12].

| Input/ancillary data | <ul><li>Malaria endemicity based on 43,187 parasite rate points in sub-Saharan Africa collected from 2000 to 2017.</li><li>Malaria Control Interventions: Insecticide-treated bednets, indoor residual spraying and effective antimalarial drug treatment.</li><li>Temperature: Daytime land surface temperature (LST), nighttime LST, delta LST and temperature suitability for *P. falciparum* transmission.</li><li>Precipitation: Magnitude, variability and seasonal rate of change in precipitation.</li><li>Land cover types.</li><li>Surface Moisture and Vector Breeding Sites: Normalized difference wetness index, Tasseled Cap wetness, Tasseled Cap brightness, potential evapotranspiration, and aridity index.</li><li>Enhanced Vegetation Index.</li><li>Slope angle, flow accumulation, and topographic wetness index.</li><li>Population density, nighttime lights data, and accessibility to cities with populations exceeding 50,000, represented as cost distance friction raster.</li></ul> |
|---|---|
| Strengths | <ul><li>Fine-grained evaluation of intervention-burden links.</li><li>Offers more detail than other studies that pooled Pf estimates by admin level.</li></ul> |
| Limitations | <ul><li>Data contribute to World Malaria Report 2017 and GBD studies, making comparisons with alternate global burden estimates challenging (as these seem to be the only alternate sources).</li></ul> |
| Uncertainty | <ul><li>Parasite rates predicted using Bayesian space-time geostatistical model.</li><li>Various co-variates with own uncertainties as model input data.</li><li>Some co-variates themselves are modeled data (e.g., malaria control intervention).</li></ul> |
| Validation | <ul><li>Results compared to two World Malaria Report 2017 and GBD studies: 2000-10 results were similar. In 2016, MAP estimated fewer cases than GBD and WMR 2017. Fatalities: MAP estimated fewer deaths than GBD 2016. MAP estimated more deaths (40.7%) than WMR 2017 due to different mortality calculation approaches.</li></ul> |

### 4.2. The Climate in the Upcoming Months Being Particularly Conductive to Mosquito Breeding, i.e., Expectations of Above-Average Precipitation

For this indicator, two input data sources were used: Historical precipitation patterns based on CHIRPS and precipitation forecasts provided by ECMWF.

Data: CHIRPS—Rainfall estimates from rain gauge and satellite observations.

### 4.2.1. Quality by Design

CHIRPS covers 50° N to 50° S and all longitudes, meaning that our ROI is fully covered, with a spatial resolution of 0.05°. The data are offered in different temporal resolutions, from which we chose monthly, and the temporal coverage ranges from 1981 to near real time. These design criteria align with our use case objectives by allowing concrete insights into the seasonal precipitation patterns of our ROI. However, CHIRPS state that their primary goal is to monitor agricultural drought [32], i.e., the absence of

precipitation, while we were interested in the presence of precipitation. While this seems like the same phenomenon, it had implications in that it introduced negative biases until the year 2000, which were effectively removed for the more recent years [66]. Figure 6 shows the combination of historical CHIRPS data with ECMWF forecast data in one integrated precipitation risk indicator.

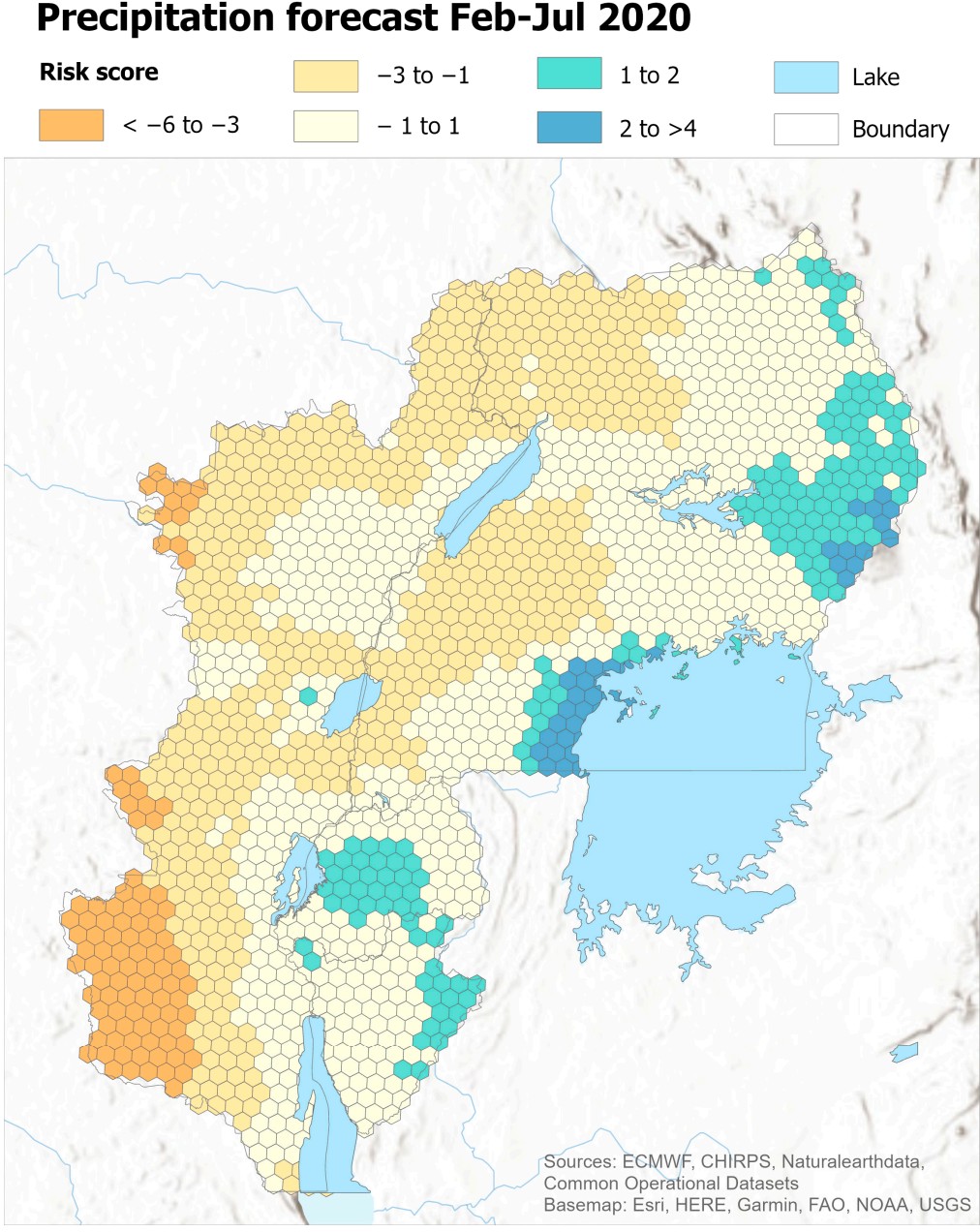

**Figure 6.** The integrated indicator that combines historical precipitation data with precipitation forecasts.

### 4.2.2. Quality of Conformance

The methodology used in generating CHIRPS data is comprehensively documented [32,67,68]. Research indicates that in regions with limited ground-based weather station coverage, and where satellite data play a more significant role, the estimations tend to be less reliable (see Table 4) [66,69–71]. Additionally, areas with complex topography also pose challenges for accurate estimations. Both sources of inaccuracy are likely present in our ROI. While we cannot precisely judge the impact of these limitations

on our ROI, our primary focus was not on precise precipitation values. We analyzed the data in a more aggregated manner, and for this purpose, we estimated that CHIRPS data serve as an adequate source for understanding long-term seasonal precipitation patterns. In comparison to other datasets with a comparable spatial coverage, they have furthermore been shown to be more reliable, and provide better insights [32].

**Table 4.** The information presented in this table comes from [32,69,72].

| | |
|---|---|
| Input/ancillary data | • Meteorological station data: From various public and private organizations worldwide [32].<br>• Satellite data: Tropical rainfall measuring mission, monthly mean geostationary infrared brightness temperatures, land surface temperature (MODIS) [32].<br>• Topographic and physiographic surfaces: Elevation and slope, 30 arc seconds [32]. |
| Strengths | • More reliable globally than comparable products [32].<br>• Provides reliable information on monthly or yearly scale [32].<br>• Estimations available with few latencies [32]. |
| Limitations | • Studies have found that it tends to over- or underestimate, especially in complex terrains [32].<br>• In areas with very few station observations, some other models perform better [32]. |
| Uncertainty | • Low coverage of ground stations means higher weighting of satellite data, increasing uncertainty [69].<br>• Higher topographical complexity leading to higher deviations [69].<br>• Studies show different results in different areas globally (see Annex I (Supplementary Materials) for details). |
| Validation | • Compared their results for Afghanistan, Colombia, Ethiopia, Mexico and the Sahel (Senegal, Burkina Faso, Mali, Niger and Chad) to high-quality gauge data obtained from the national meteorological agencies of the regions [72]. |

Data: ECMWF—Seasonal precipitation forecast.

### 4.2.3. Quality by Design

The ECMWF's seasonal precipitation forecast is a global dataset derived from the SEAS5 model, featuring a spatial resolution of approximately $1° \times 1°$ (about 111 km). It provides precipitation anomaly projections up to six months in advance, offering valuable early warning capabilities. However, the dataset's resolution is notably coarser than our target hexagonal grid, resulting in abrupt boundaries within our ROI (see Figure 4). These sharp edges may not accurately represent actual precipitation patterns. The forecast data are primarily intended for a broader-scale analysis than our ROI. Consequently, it remains uncertain how adequate this dataset is for our specific use case, given the differences in design and scale.

### 4.2.4. Quality of Conformance

The quality evaluation showed that the ECMWF dataset comes with user guides and several other quality assessment criteria, as suggested by [41]. However, not all of the criteria have been documented yet. As shown in Table 5, the precipitation forecasts have been found to be most reliable for tropical ocean areas, with land forecasts presenting challenges [73]. Despite limited utility in local seasonal rainfall predictions, average values for tropical regions show significant skill, and are crucial for extratropical predictability. The SEAS5 sea-surface temperature (SST) forecast for El Niño—Southern Oscillation (ENSO) prediction exhibited high to very high skill levels across the Pacific, regardless of lead times. This is valuable because ENSO is associated with higher malaria risk in parts of Africa [74]. It is important to note that seasonal forecast systems exhibit biases with varying spatial patterns that tend to increase as the forecast time lengthens. While it is difficult for us to estimate this dataset's adequacy for the use case, in particular its use as part of a composite indicator, we acknowledge its relevance in malaria intervention planning. It may be more prudent to ensure consistent monitoring of this dataset by MSF staff on a broader continental to global scale, rather than limiting the focus to the region of interest.

**Table 5.** The information in this table comes from the C3S Climate Data Store Website [73] and the C3S Knowledge Base [75].

| Input/ancillary data | • Input data provided by various meteorological offices globally [75]. |
|---|---|
| Strengths | • See limitations [73]. |
| Limitations | • Seasonal forecast quality is generally better over the (tropical) oceans than over land [73].<br>• SEAS5 SST forecast skill for ENSO prediction is generally high to very high across the Pacific at all lead times [73].<br>• Precipitation is best predicted over parts of the tropical oceans, while seasonal prediction for rainfall over land is, with some exceptions, challenging [73].<br>• Although seasonal local rainfall predictions are often not useful, average values on tropical regions have significant skill and play a crucial role for extratropical predictability [73]. |
| Uncertainty | • Seasonal forecast systems have biases with spatial heterogeneity that grow with forecast time, stemming from different biases of the model [73]. |
| Validation | • Scientific evaluation and validation carried out as part of the implementation of SEAS5, reported in [36,76]. |

### 4.3. Limited Access to Healthcare—Walking Time to Closest Healthcare Facility and Population per Healthcare Service Area

Limited healthcare access is represented by two indicators. The first relies on the "Walking Only Travel Time to Nearest Healthcare Facility without Access to Motorized Transport" dataset [31], which was built upon healthcare facility location data by [34]. Due to the significance of this underlying healthcare facility dataset, we also evaluated it, with the results documented in Table 6. However, the written text focuses on the "Walking Only Travel Time to Nearest Healthcare Facility without Access to Motorized Transport" dataset. The second indicator is built on the same two data sources, complemented by the inclusion of a WorldPop dataset.

Data: Walking-only travel time to nearest healthcare facility without access to motorized transport.

**Table 6.** The details for the dataset "A spatial database of health facilities managed by the public health sector in sub-Saharan Africa". Information presented is taken from [31,34].

| Input/ancillary data | • 93 different sources: for example, Ministries of health (MoH), UN bodies, non-governmental organizations, personal communications [34]. |
|---|---|
| Strengths | • Includes public facilities and private-not-for-profit health facilities [34].<br>• Duplicates removed by authors [34]. |
| Limitations | • Excludes private-for-profit health facilities, government facilities (e.g., prisons), blood transfusion centers, HIV voluntary counseling and testing centers, maternity and nursing homes, family planning clinics and specialist facilities (e.g., dental); spatial locations not universally documented across national health facilities listings [34].<br>• Definitions of facility types vary between countries [34].<br>• Completeness and accuracy of facilities vary by country [31].<br>• Facility may be open but understaffed or closed seasonally or permanently [31].<br>• Not all facilities offer the same services [31].<br>• Focus is on geographically fixed facilities, no mobile or temporary clinics (important for facilities in remote areas) [31]. |
| Uncertainty | • Completeness of input data varies by country [34].<br>• Possible geolocation uncertainty for facilities where no coordinates were available (in those instances, different manual techniques were used to locate the facilities) [34]. |
| Validation | • Visual inspection in Google Earth [34].<br>• Checked whether health facilities are in correct administrative zone and on land [34].<br>• Country-specific definitions of service levels compared with existing databases [34].<br>• Number of health facilities at each level compared to health sector strategic plans (HSSP) data, revealing mostly similar numbers and occasional discrepancies (often due to underreporting of NGO facilities and temporal data differences) [34]. |

### 4.3.1. Quality by Design

The dataset estimates travel times in minutes from every location to the nearest healthcare facility by walking, with a spatial resolution of 1 km (see Figure 7). A limitation is that the healthcare facility data were last curated in mid-2019 and, to the best of our knowledge, no updated version is yet available. Considering potential changes that happened since then, this dataset may be outdated for future applications, although new initiatives are underway [77,78].

### 4.3.2. Quality of Conformance

The methodology and input sources are described in several journal publications [31,34,56]. The adequacy of the walking time hinges on the completeness and location accuracy of the healthcare facilities. This, however, varies by country, and it is uncertain to us how each country performs (see Table 6). However, the healthcare facility data by [34] are still considered to be the most comprehensive dataset currently available [79]. The walking time itself is to be considered an estimate of potential, rather than actual travel times, and it does not account for possible differences in travel time due to seasonality, age or health status (see Table 7). We found the resulting indicator sufficient for providing a broad overview of regions with limited healthcare accessibility. However, given the uncertainties that we identified, and after comparing it with an internally used MSF healthcare facility database, we would advise against using this data source for operational planning.

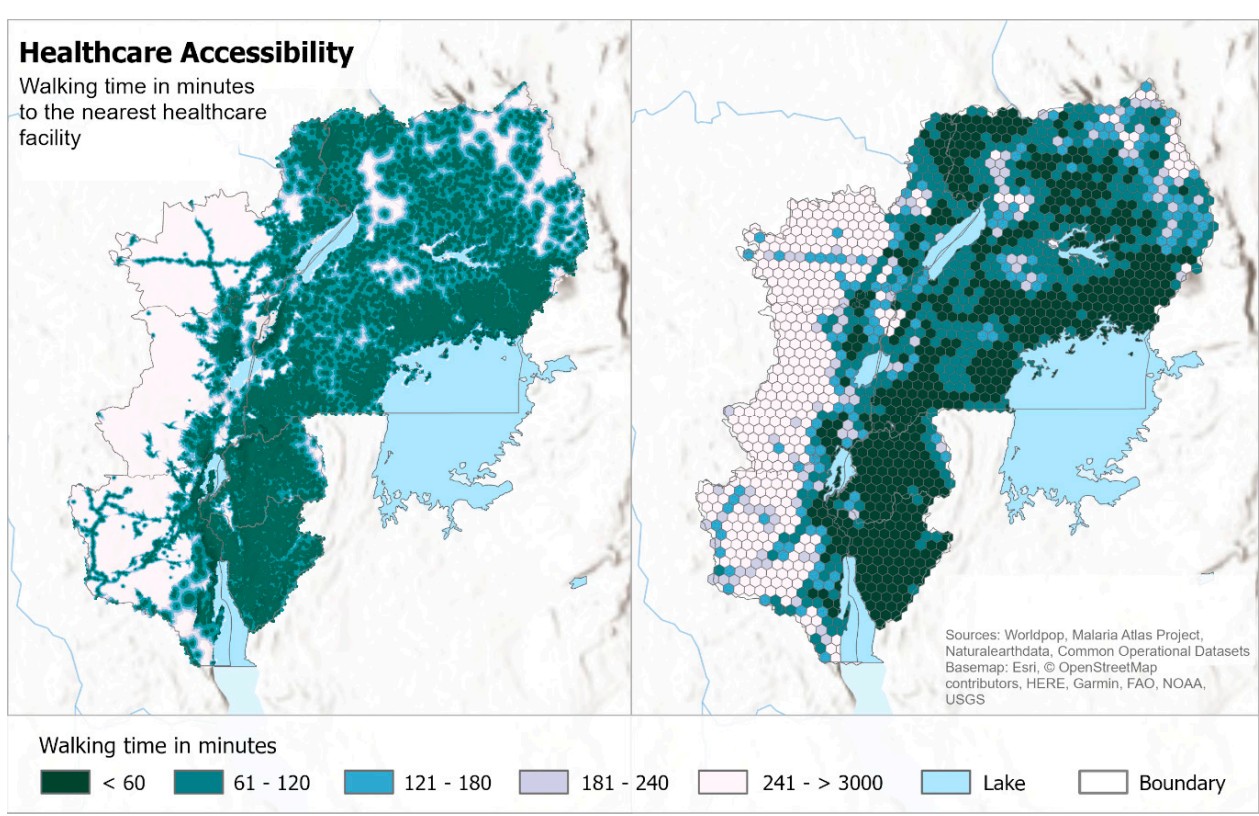

**Figure 7.** The map on the left displays the original data as provided by the MAP. The map on the right shows the same data after we integrated them into our hexagonal reporting grid.

### 4.3.3. Quality by Design

The data contain population figures per pixel, with country totals adjusted to align with official United Nations population estimates [80]. We opted for WorldPop's "unconstrained" dataset, which assigns population values to all land grid cells, in contrast to their "constrained" datasets, where population is allocated exclusively to areas recognized as buildings and settlements [81]. This choice was made for its enhanced accuracy in regions

where satellite-based settlement mapping is uncertain, particularly for small rural settlements [81]. In terms of spatial and temporal coverage, global data are available for each year from 2000 to 2020. We chose the data that have a resolution of 1 km. Furthermore, we chose the UN-adjusted version of the data because it is recommended to use the adjusted version in areas where no recent census data are available, which is the case in our ROI. See Figure 8 for an overlay of the population counts with healthcare service areas.

**Table 7.** The details for the dataset "Walking-only Travel Time to Nearest Healthcare Facility without Access to Motorized Transport". Information presented is taken from [31].

| Input/ancillary data | • Healthcare facilities, see Table 6 or [34].<br>• Walking time: OSM and Google, roads, railways, waterways, land types and associated travel times, slope angle and atmospheric density (Tobler Hiking Function) [31]. |
|---|---|
| Strengths | • First global-scale, high-resolution maps of facility accessibility [31].<br>• Friction surface and travel time mapping code freely provided, allowing for producing custom maps of travel time [31]. |
| Limitations | • Variability in travel times not accounted for (e.g., due to seasonal conditions, age or health status) [31].<br>• Travel time is merely an estimate of potential [31]. |
| Uncertainty | • Uncertainty mostly comes from uncertainty related to healthcare facility data, which vary by country [31]. |
| Validation | • Results compared to Google Maps [31].<br>• Travel times on average ± 15.8 min of those from the alternative source [31].<br>• Spatial variability in model accuracy, some areas prone to overestimates and others to underestimates [31]. |

Data: WorldPop: Population counts—unconstrained individual countries 2020, UN adjusted.

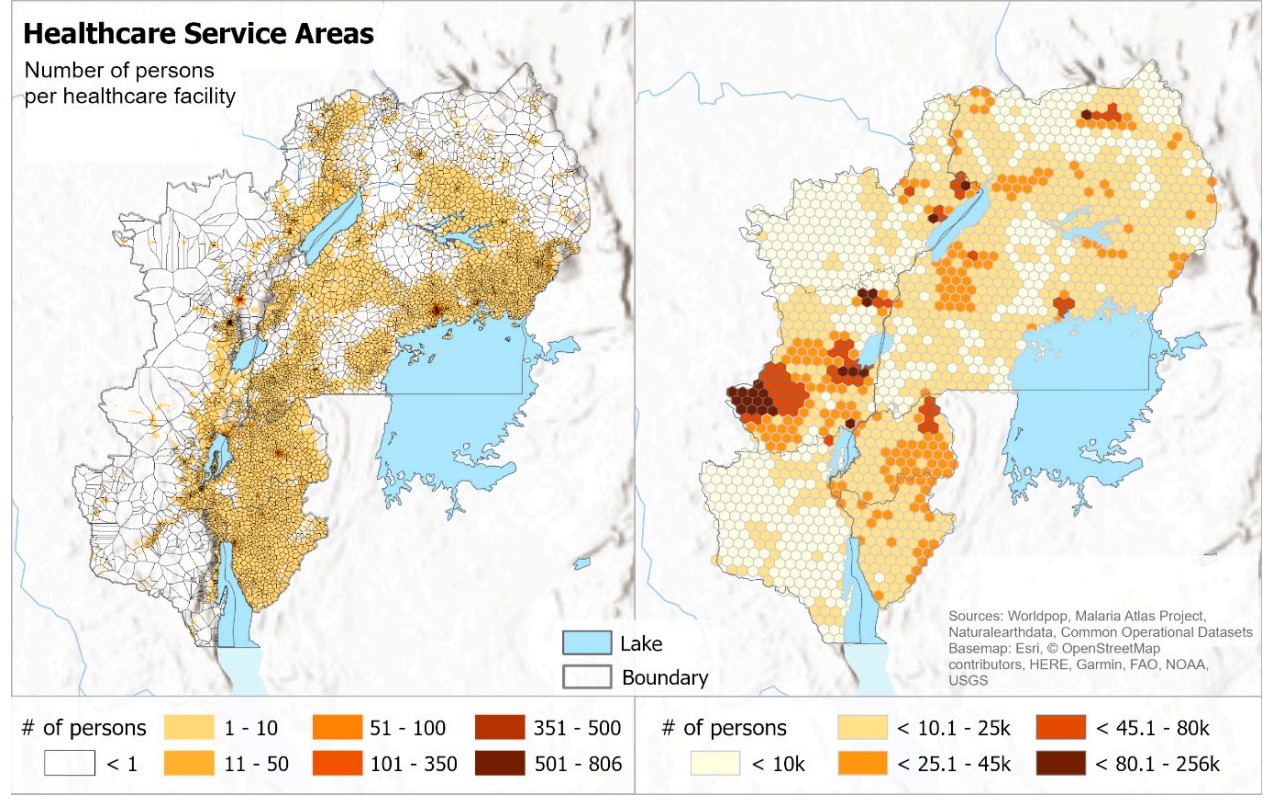

**Figure 8.** The map on the left displays the healthcare service areas we derived from the accessibility surface, overlaid with WorldPop population estimates. The map on the right shows the number (#) of people per individual healthcare facility, integrated into our grid.

4.3.4. Quality of Conformance

WorldPop has documented their methods in a series of journal publications [33,82]. However, our task of gathering all relevant information was challenging, given its dispersion across multiple sources, including WorldPop, UN reports and the previously mentioned publications.

To generate population counts, WorldPop used national UN population estimates from two specific points in time. These estimates could be derived from either census data or estimations [80]. Utilizing these two reference points, they calculated population estimates for each year from 2000 to 2020. Then, the national statistics were spatially disaggregated based on various input data sources, with their own imperfections (see Table 8). The most recent of these input data date back to 2017, with road data, for example, sourced from OpenStreetMap (OSM) in 2016. We compared the number of OSM road elements for our ROI in January 2017 and October 2023, and found that the number of elements labeled as roads increased by 841%. Given the multiple input datasets, it is uncertain how adequate the population estimates are. Furthermore, the referenced dataset was found to underestimate populations in slums and high-density urban areas in Namibia [25]. It remains uncertain how well WorldPop data can reflect larger-scale migration movements, let alone rapid displacement, which play a role in our context.

**Table 8.** The details for the dataset "Population Count—Unconstrained individual countries 2020 UN adjusted", created by WorldPop. Information presented here is taken from [82] and different WorldPop sub-websites [57,81,83].

| Input/ancillary data | <ul><li>UN population estimates on admin 0 level [57].</li><li>Land cover [82].</li><li>Raster: Annual NPP 2010, lights at night, mean temperature 1950–2000, mean precipitation 1950–2000, elevation, slope [82].</li><li>Vector: Distances to roads, distances to rivers/streams, generic populated places, water bodies, protected areas, canals, communities, district seats, cities, hamlets, villages, suburbs, towns, populated points, railways, generic health facilities, health clinics, dispensaries, hospitals, schools, settlement points, built land cover [82].</li></ul> |
|---|---|
| Strengths | <ul><li>The datasets are suitable where the accuracy of the satellite-based mapping of settlements is uncertain, especially in the detection of small rural settlements. The global multi-temporal nature of the datasets also makes these data the best option for historical or change analyses [83].</li><li>Multi-temporal global data available for each year, 2000–2020 [57].</li></ul> |
| Limitations | <ul><li>Method produces a non-zero allocation of population to all land grid cells, resulting in misallocations of population to uninhabited areas, and underestimates urban population in some areas [83].</li></ul> |
| Uncertainty | <ul><li>Proxies used to determine likelihood of population occurrence (e.g., occurrence of healthcare facilities, night lights, distance to roads, etc.) [82].</li><li>Estimation method only suitable for stationary communities [82].</li><li>Reliance on auxiliary data creates a dependency on and reproduction of uncertainties in source datasets [82].</li><li>Comparison data (e.g., census), relatively old in parts (e.g., 1999–2008 for Burundi) [80].</li><li>Random forest classification algorithm can predict numbers beyond the maxima in the training data [82].</li></ul> |
| Validation | <ul><li>Comparison with census data of each respective country used as validation (a challenge, given that census designs differ by country).</li></ul> |

*4.4. Ongoing Conflicts*

4.4.1. Quality by Design

The armed conflict locations and events database (ACLED) [84] dataset provides real-time, event-based information on global political violence, demonstrations and related non-violent events. It includes event type, actors, location, date and other details, following established methodologies for weekly publication [85,86].

The data support the objective of identifying conflict-affected regions within the ROI, offering city or village-level precision, which aligns with our targeted scale. Input and output data are shown in Figure 9.

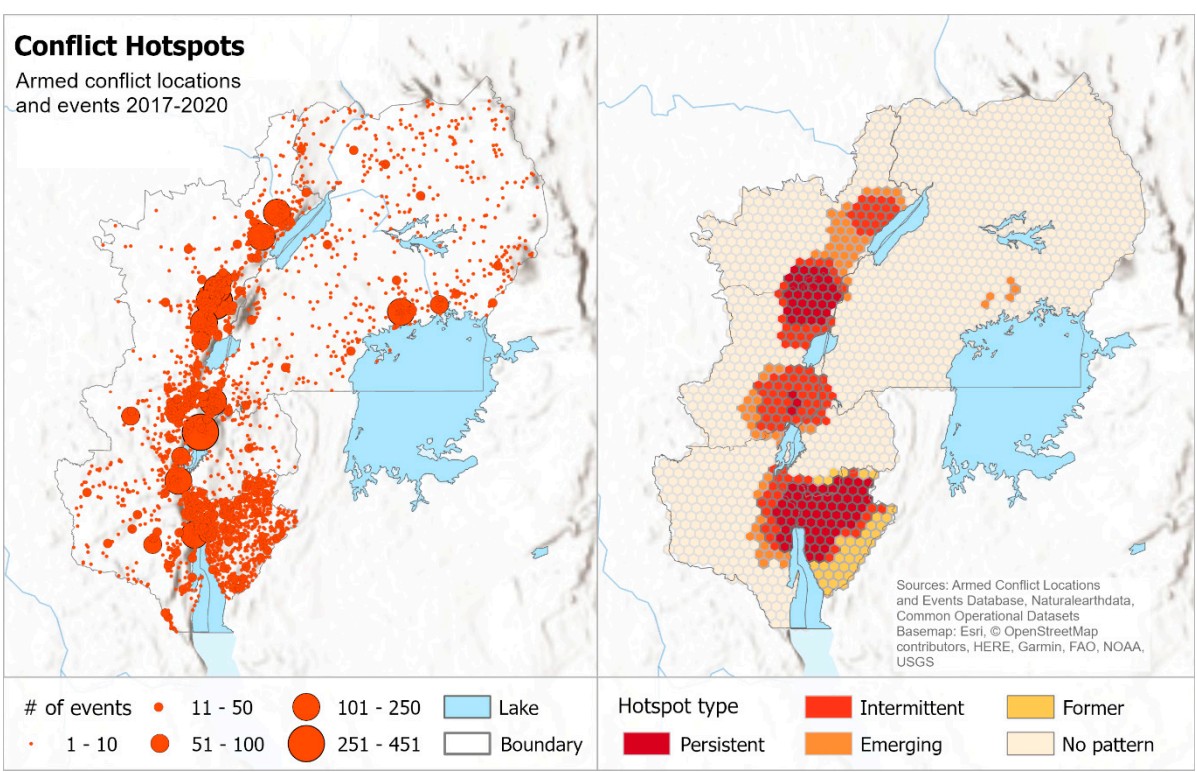

**Figure 9.** The map on the left displays the original data as provided by ACLED. The map on the right shows our hexagonal grid classification into different types of hotspots based on events' locations and timing.

### 4.4.2. Quality of Conformance

The ACLED methodology is comprehensively documented in the resource library, providing insights into its strengths, limitations, sources of uncertainty and attempts to validate the data [86] (see Table 9). However, assessing the database's reliability in a specific area presents challenges. In regions under the control of militia or rebellious groups, as is the case in parts of the ROI, these groups may be the only source of information [35]. These sources are not necessarily impartial in their reporting, and neutral entities lack oversight in these areas. However, the uncertainties introduced by potentially biased sources are likely minimized in significance due to our rather coarse temporal and spatial resolution objectives. In general, the in-depth evaluation of the ACLED data underlined its leading role as a conflict database over others: it outperforms comparable databases in terms of data collection and oversight, inclusion, coverage and classification, usability and transparency, and sourcing [35]. The described methodology indicates a robust strategy for data collection and systematization.

**Table 9.** The information shown here describes the ACLED database.

| Input/ancillary data | • Traditional Media: Subnational, national, regional and international outlets following journalistic verification principles [87].<br>• Reports: From NGOs, international institutions, human rights organizations, investigative journalism groups, and, in specific situations, ministries of defense, armed groups, NATO, etc. [87].<br>• Local Partner Data: Collected from local-level observatories and activists [87].<br>• New Media (verified): Includes Twitter, Telegram, WhatsApp, with direct source contact or alternative verification methods—no crowdsourcing or web scraping involved [87]. |
|---|---|

**Table 9.** *Cont.*

| | |
|---|---|
| Strengths | • Uses diverse, multilingual sources. Prioritizes local and sub-national media because traditional sources can create biases, favoring safer regions and sensational events, while neglecting smaller or prolonged conflicts [87]. |
| Limitations | • Data not intended for day-to-day safety monitoring, lacking specific event times and street-level details. City/village-level data are provided, occasionally disaggregated to neighborhoods in large cities [88]. |
| Uncertainty | • Information sources may exhibit bias, such as when an involved actor is the sole reporter in a given area [87]. Each event includes a "GEO_PRECISION" column, denoting location uncertainty with a numeric code [89]. |
| Validation | • ACLED does not independently verify events but collaborates with local partners, which are carefully evaluated [35,89]. |

## 5. Discussion

### 5.1. The Evaluation Framework

The increasing availability of geospatial data is enabling risk modelers to rely increasingly on openly available datasets created by specialized research groups [90–92]. This trend reflects the typical progression of a new field, where individuals must initially handle all aspects of the research; however, as the field evolves, it becomes segmented into more specialized areas with their own experts. In our capacity as practitioners of indicator-based risk assessments, our role evolved from undertaking the entire data modeling process to focusing on effectively conceptualizing and selecting the most adequate data for a given use case. In essence, we became "data pharmacists" who engage with stakeholders to identify relevant problems, conduct research on available data, determine the most adequate options based on the data's strengths and limitations, and navigate the challenges (including error propagation and unwanted "drug interaction") associated with their use. However, to be able to provide such an estimation, we needed a standardized set of data characteristics that we could refer to. The data evaluation framework is a first step in the direction of systematically assessing and comparing various aspects of geospatial data, and aligns with approaches in related domains [93]. Differentiating between "quality by design" and "quality of conformance" provides a means of discussing and expressing two distinct dimensions of adequacy or quality. Future research may aim to quantitatively measure evaluation criteria alongside their qualitative descriptions. Nevertheless, this task remains challenging due to limited options for validating quality of conformance criteria in the presence of known and unknown uncertainties.

Evaluating the adequacy of a dataset for a particular use case remains a time-intensive task that demands research skills, and a profound understanding of modeling and validation procedures. These requirements can exceed the capabilities of individual risk modelers, particularly when dealing with multi-source data with various associated uncertainties, which is why dedicated research groups are required. However, particularly with the rise of machine learning and AI applications, the importance of understanding the input data and potential biases cannot be stressed enough. This prompts a critical question for the future: Who bears the responsibility for providing the information necessary for evaluating data adequacy—the data producer or the user? The producer should transparently provide all necessary/possible information for the user to estimate data adequacy, but the user is obligated to gather and judge this information. Hence, future efforts should aim to establish a standardized set of contextual information provided by the data producer that is easily accessible in one place by the user.

### 5.2. Use Case

The data we gathered proved to be somewhat useful in developing the envisioned early warning malaria tool, albeit with certain limitations.

Having the MAP malaria incidence data with a yearly temporal resolution was valuable for identifying general trends. However, monthly disaggregated numbers indicating

the general malaria pattern throughout the year would be necessary to plan medical interventions ahead of the peak transmission period.

Interpreting precipitation forecasts posed challenges for us as non-climatologists. This is unfortunate, as these forecasts hold significant value across various applications and have piqued the interest of MSF staff. However, efforts for more educational resources for non-experts are currently being developed [94,95]. While the applied methodology effectively preserved spatial detail and offered insights into the ROI's precipitation patterns—an aspect that, to our best knowledge, was previously unavailable—the approach to defining the rainy season and calculating the risk score was somewhat generic. As a potential next step, collaborating with specialists could refine this indicator.

The healthcare facilities dataset exhibits limitations due to potential outdatedness and incompleteness, especially in troubled areas with significant MSF activity. The accessibility surface shares the limitations with the healthcare facilities data, and has its own limitations in areas where, for example, the OSM completeness levels in the past were low. Still, we consider these resources valuable for the assessment to show the general pattern of healthcare accessibility in the ROI.

The estimation of individuals seeking healthcare in the same facility, based on World-Pop data, falls short in reflecting the population in rapidly established internally displaced persons or refugee camps—a relevant aspect in our context.

While the ACLED data were adequate to highlight conflict hotspots, we had initially planned another indicator that informs about locations that people seek refuge in. The International Organization for Migration Displacement Tracking Matrix (IOM DTM) offered recent and detailed refugee and internally displaced persons (IDP) statistics for the three provinces in DRC (IOM 2022; 2023a; 2023b). However, for Burundi, Rwanda, and Uganda, no comparable data were available, leading us to discard the indicator for the time being.

Overall, the openly available geospatial data demonstrated high quality. Substantial effort has been invested in modeling over the past years, and ongoing initiatives continue to drive this process further.

We successfully obtained freely available geospatial data for the majority of our desired indicators, meeting our spatial and temporal resolution. For applications in public-health contexts, the availability of data is becoming less of a bottleneck, while the ability to evaluate their adequacy is becoming increasingly important.

Since we cannot definitively determine the adequacy of the data due to uncertainties, it is challenging to decide whether they should inform operational planning on the MSF side. While several datasets offer fine resolutions, questions remain about the accuracy of estimations at the cell level. While we are uncertain about their applicability to operational planning, we would recommend that MSF and other organizations remain aware of these datasets and their future development for awareness-raising and advocacy purposes. Our data assessment was acknowledged by MSF; individual datasets were integrated into their database, and the evaluation framework has been applied in a simplified form. Currently, we have intentionally avoided presenting the individual indicators as an integrated risk surface. Creating such a surface would necessitate a descriptive text to interpret the resulting spatial patterns. We believe this could falsely imply certainty and detract from the primary message of our research.

Looking ahead, there is optimism that in the future, we may achieve a more robust and certain assessment for planning sub-national and targeted interventions. As data sources and methodologies continue to evolve, our aspirations for accurate and effective planning will likely become more attainable. The interest sparked among the MSF staff underscores the practical significance by addressing genuine needs within the humanitarian community. The data adequacy evaluation framework provides a set of criteria that risk modelers should consider before deciding to use a dataset.

## 6. Conclusions

In the realm of public health and humanitarian aid, relying on open geospatial data for spatial risk assessments often raises concerns about data quality and adequacy. To address this challenge, we introduced a systematic data evaluation framework that emphasizes "quality by design" and "quality of conformance". As risk modelers, we took on the role of "data pharmacists" who collaborate with stakeholders to diagnose information deficiencies, seeking to find the right "cure". We explored various datasets, determining their adequacy. This selection process carefully balances the potential risks posed by data limitations with the "healing" qualities found in the data's strengths. Through an applied use case with MSF, we evaluated a range of data sources for indicator data, applying our framework to assess their suitability for operational intervention planning. While data availability and contextual information are generally provided, determining their adequacy definitively for humanitarian intervention planning remains challenging. This is due to potential data inaccuracies, incompleteness or outdatedness that are difficult to quantify, particularly in modeled data with complex input covariates. From the user's/risk modeler's perspective, the framework may aid in planning a systematic approach to data adequacy evaluation. From the data provider's perspective, it shows a real-world application in which their data are being considered for use. In line with the "do no harm" principle, it is crucial not to misrepresent certainty in our assessments, especially in human-centered risk assessments. Our foremost concern is the well-being of MSF beneficiaries, and misrepresenting certainty could be a disservice.

**Supplementary Materials:** The following supporting information can be downloaded at: https://www.mdpi.com/article/10.3390/ijgi13020033/s1, Supplementary information is provided in Annex I, which contains the Data Evaluation Framework applied to all datasets evaluated for this project.

**Author Contributions:** Conceptualization, Linda Petutschnig, Thomas Clemen, Ulfia Clemen and Stefan Lang; methodology, Linda Petutschnig and Stefan Lang; software and code, Linda Petutschnig; validation, Linda Petutschnig; formal analysis, Linda Petutschnig and E. Sophia Klaußner; investigation, Linda Petutschnig and E. Sophia Klaußner; resources, Stefan Lang; data curation, Linda Petutschnig; writing—original draft preparation, Linda Petutschnig, E. Sophia Klaußner, Thomas Clemen, Ulfia Clemen and Stefan Lang; writing—review and editing, Stefan Lang, E. Sophia Klaußner and Thomas Clemen; visualization, Linda Petutschnig; supervision, Stefan Lang and Thomas Clemen; project administration, Stefan Lang; funding acquisition, Stefan Lang. All authors have read and agreed to the published version of the manuscript.

**Funding:** This research was funded by the Austrian Federal Ministry of Labour and Economy, the National Foundation for Research, Technology and Development, the Christian Doppler Research Association, and Médecins Sans Frontières (MSF) Austria. Open access publication supported by the University of Salzburg Publication Fund.

**Data Availability Statement:** All data created for the use case are available on Figshare, https://doi.org/10.6084/m9.figshare.24434812 (accessed on 3 November 2023). The developed analysis workflow coded in R can be found on GitHub (https://github.com/Menkli/malaria_risk, accessed on 3 November 2023).

**Acknowledgments:** We would like to express our gratitude to the MSF staff who contributed with their expert knowledge.

**Conflicts of Interest:** The authors declare no conflicts of interest.

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
