# Peer review of "Evaluating Geospatial Data Adequacy for Integrated Risk Assessments: A Malaria Risk Use Case"

_ijgi, doi:10.3390/ijgi13020033_

Round 1

Reviewer 1 Report

Comments and Suggestions for Authors

Refereeing for IJGI Paper Evaluating Geospatial Data Adequacy for Integrated Data Risk Assessments A Malaria Case Study

This is a truly excellent paper and this journal would be fortunate to publish it since this paper is likely to be useful to a great many researchers doing geospatial health care research in Africa (and therefore likely to be well cited). It is also well-written and a pleasure to read unlike so many of the papers that I referee where I find myself constantly re-writing the excruciating English and dreadful grammar. I could not find any “typos” at all – except one: line 63 constrains should be constraints. This is not only a well written paper but it is well produced and presented with an outstanding set of maps and tables that are integral to the text.

The paper presents a case study of malaria but it is, of course, a model for so many other tropical diseases that have recently been analyzed in academic papers including, for example, dengue, zika, rift valley fever among so many others. It is impressive that the authors have had the collaboration of Medicins Sans Frontieres (MSF) who are globally celebrated and are arguably the best-known humanitarian aid organization. Collaboration with MSF does much to guarantee the quality and usefulness of this research paper.

Regarding the Abstract: The abstract presents an effective and lucid summary of the whole paper. However, to increase its usefulness please spell out all the acronyms.

Comments on the Paper:

1. Introduction: The “domain agnostic” approach guarantees the usefulness of the research in other domains/diseases. The references are extensive and necessary and helpful. However, all references that are to websites (generally those not in academic journals) should come with a  URL link so the reader can find the data/code etc. that is referred to (if this journal has a policy against this then I suggest you change that policy). I was surprised that there were no references to the work of Simon Hay and/or Janey Messina since their highly cited research in the best journals has extensively covered geospatial research on the distribution and prediction of African diseases.  Here is a link to Dr. Messina’s relevant research: https://scholar.google.ca/citations?user=OpURMy0AAAAJ&hl=en&oi=sra

And citations of two relevant papers:

Messina, J.P., Pigott, D.M., Golding, N., Duda, K.A., Brownstein, J.S., Weiss, D.J., Gibson, H., Robinson, T.P., Gilbert, M., William Wint, G.R. and Nuttall, P.A., 2015. The global distribution of Crimean-Congo hemorrhagic fever. Transactions of the Royal Society of Tropical Medicine and Hygiene, 109(8), pp.503-513.

Howes, R.E., Dewi, M., Piel, F.B., Monteiro, W.M., Battle, K.E., Messina, J.P., Sakuntabhai, A., Satyagraha, A.W., Williams, T.N., Baird, J.K. and Hay, S.I., 2013. Spatial distribution of G6PD deficiency variants across malaria-endemic regions. Malaria journal, 12(1), pp.1-15.

2. Problem Statement: I enjoyed the discussion of FAIR (findable, accessible, interoperable, reusable) data sharing principles. The authors should also mention the concepts of replication and reproducibility (R&R) as part of this discussion. This has recently been the topic of a Forum in the Annals of the Association of American Geographers and one of these papers also refers to modeling of Rift Valley Fever in Africa. It also discusses the use of Jupyter Notebooks to enhance principles such as FAIR and the R&R of research (Waters, N., 2021. Motivations and methods for replication in geography: Working with data streams. Annals of the American Association of Geographers, 111(5), pp.1291-1299.) Please drop reference #13: it is easy to find on Google Scholar but is a rather unpleasant example and only “alleged” while references 14 and 15 are more than adequate. Reference 36 to “Nordic Actor-Oriented Climate Adaptation Research” is a bit of a stretch when this paper is so heavily concerned with Africa but it does emphasize support from “other domains”. I appreciated the explicit explanation of the paper’s two target audiences at the end of section 2.

3. Background and Methods: The definition of adequacy is really useful. However, I really don’t like some of the words used. For example, I think it should be “Quality of Design” (not in) and “Quality of Conformance” should be spelled out at the start i.e. “Quality of Conforming to Accuracy and Completeness”. This would be more transparent. However, it would kind of mess up the whole paper and since Table 1 is excellent and does an impressive job of explaining all the details of the evaluation criteria, let’s leave it at is.

An additional and more up to date reference (than #59) on Discrete Global Grid Systems is Peterson’s other paper which is in the special issue of Cartographica on the topic: Purss, M.B., Peterson, P.R., Strobl, P., Dow, C., Sabeur, Z.A., Gibb, R.G. and Ben, J., 2019. Datacubes: A discrete global grid systems perspective. Cartographica: The International Journal for Geographic Information and Geovisualization, 54(1), pp.63-71.

4. Results:

Title of Table 3 the word “come” should be “comes”. Apart from that Table 3 is another exceptional summary of the issues discussed in the text. Well done.

Line 350: please be specific and describe the “implications”.

Line 358: “Research indicates….” Please cite references to this research.

Section 4.3.2 Quality of Conformance: Reference 54 “Global Map of Travel time to Cities to Assess inequalities in Accessibility in 2015”. Published in 2015 in Nature – This Map would appear to have completely omitted Vancouver…. ☹

Section 4.3.4 is an excellent example of the meticulous research that was conducted in this paper. Well done.

Table 8: Under validation please mention that it is a major issue that censuses from different countries differ in many ways and thus conformity to a single ideal is a challenge.

Table 9: Validation row in the Table mentions ACLED “collaborates with local partnerships”. Can it be mentioned how these partnerships vary?

5. Discussion: I really like the “pharmacist” metaphor (although it took me a moment to get my head around it). Error propagation is always a concern in geospatial work with multiple layers/data sets. Prescribing multiple drugs to a single patient is an issue because of possible drug interactions. Can error propagation and drug interaction be part of the metaphor – or is that what the paper had in mind all the time?

Lines 527-529 Requirements of the research are stated to exceed the capabilities of individual modelers – that, of course, is why this type of work has to be carried out by teams of researchers with different capabilities.

Lines 575-580 Did you get specific feedback from MSF. If so please include it in the report.

References 85-88 are most useful.

Summary. This is an outstanding paper. It could be published as is but hopefully the above suggestions which are easy to make will be useful.

Reviewer 2 Report

Comments and Suggestions for Authors

An excellent research work - timely, highly interesting for the international research community, and helpful for local authorities.

Reviewer 3 Report

Comments and Suggestions for Authors

Interesting case study on the usefulness of data for a specific purpose.

The suggested framework of quality of design / quality of conformance seems a useful characterization.

Some comments regarding the specific examples:

-          While these two dimensions are important, the relevance of the data depends on the quality of the knowledge regarding their possible impact on the variable of interest, in this case malaria incidence. While the authors provide a candid analysis, indicating how the data used (eg: reduction in malaria instead of malaria seasonality) may be different from the data initially planned, it is important that how useful each of these datasets are for this specific purpose depends on our knowledge of the contributors of malaria. This is beyond this paper, but it permeates its content. How can we be sure that the specific indicators chosen are the relevant ones for our specific purpose? I think the framework of quality of conformance should include one more section dealing with these aspects. Is this the data we need for this particular application? Based on what evidence? And what principle? Note that this has not much to do with the data itself, rather its adequacy for a specific purpose.

-          In this line, the article does not sum up the analysis on how all the information is combined to detect regions at high risk. It seems that was the guiding principle, how was all the information combined, if it was, or how could it be.

-          Regarding both the quality by design and quality of conformance, there is, in my opinion, little attention paid to the underlying primary data. Note that one thing is the resolution provided and another the resolution of the original source. In knowing the uncertainty of the estimates, for instance, this depends on the available of primary data such as censuses (with or without microdata), health surveys, malaria indicators surveys, … and their respective samples sizes, or the number of weather stations. While it is clear there is a ROI here, the quality will be different in different parts of the ROI based on the primary data availability, and all this might not be apparent from the data. Take a look at your “uncertainty” in table 3. You mention: “Parasite rates predicted using Bayesian space-time geostatistical model. • Various co-variates with own uncertainties as model input data. • Some co-variates themselves are modelled data (e.g., malaria control intervention).” Maybe it is not so interesting what the modelling approach was, but rather whether a measure of uncertainty is given for each data point, and whether these measures suggest that uncertainty is homogeneous or not. I think this is a different take on uncertainty, but it seems to me it is the one relevant to the user. Ultimately, the uncertainty will depend on the primary data availability as commented above. I think this should be reflected somewhere, probably in the “input/ancillary data” (now it does not, probably because all the data from MAP are put in one table, and probably a different table would be needed for each indicator.
